# Cutting the Root of Hallucination: Structural Trimming for Vulnerability Mitigation in Code LLMs

**Yage Zhang**
CISPA Helmholtz Center for Information Security
yage.zhang@cispa.de

## Abstract

We introduce a structural perspective on hallucinations in code-generating language models, framing them as causality anchors in syntax graphs that trigger cascading semantic errors and latent security flaws. This work is the first to systematically connect code hallucinations with vulnerability risks, offering a unified conceptual and practical framework to address them. At the heart of our approach is the notion of hallucination anchors, localized subtrees in the abstract syntax tree (AST) that serve as root causes of defective logic. We propose Structural Trimming (ST), a targeted mitigation method that removes these anchors while preserving functional semantics. To anticipate the effect of trimming, we introduce the Compositional Structural Hallucination Score (CSHS), which quantifies the likelihood that pruning will improve robustness. By grounding error reduction in the syntax graph itself, our method reframes hallucination mitigation as a structured intervention process interpretable, generalizable, and actionable.

## 1 Introduction

Large Language Models (LLMs) have significantly advanced automated code generation Jiang et al. (2024a); Liu et al. (2023a); Thakur et al. (2024), achieving state-of-the-art results across tasks such as autocompletion, debugging, and synthesis Chen et al. (2021); Nijkamp et al. (2023); Li et al. (2023); Huynh & Lin (2025). However, with increasing capabilities comes a growing concern: LLMs frequently produce hallucinated code plausible yet incorrect outputs that may violate functional requirements or embed latent security risks Ji et al. (2023b); Tambon et al. (2025). While prior work has predominantly addressed syntactic accuracy or model calibration, the security implications of hallucinations remain underexplored.

In this paper, we posit that hallucinations in code are not merely transient generation errors, they constitute structured precursors to real-world vulnerabilities. Unlike superficial mistakes (e.g., typos or API misuse), hallucinated variables, conditions, or control flows may propagate through the AST, forming entangled symbolic dependencies that are difficult to detect or rectify post hoc. These artifacts often persist across abstraction layers, misleading developers and enabling logic flaws.

We pursue this perspective through three research questions:

**RQ1.** How does hallucination severity relate to vulnerability risk? Can this relationship be modeled and predicted?

**RQ2.** Do hallucinations follow detectable structural patterns?

**RQ3.** Can we design effective structure-aware repair methods to reduce risk while preserving semantics?

To answer these, we construct a large-scale analysis pipeline for hallucination-driven risk assessment. Using MBPP Austin et al. (2021a), HumanEval Chen et al. (2021), and structurally annotated examples from Collu-Bench, we collect and label 7,233 hallucinated samples across four LLMs (GPT-4o, DeepSeek-Coder-1.3B, CodeLlama-7B, Gemini-2.0-flash) over

diverse tasks. We introduce a taxonomy of hallucination behaviors e.g., *persistent*, *early-cut*, *recurrent*, and *oscillating*, inspired by prior propagation typologies Ji et al. (2023b). Each sample is assigned a deviation score ($D$) and a vulnerability estimate ($V$), based on execution feedback and error severity.

To mitigate the security impact, we propose Structural Trimming (ST), a post-hoc repair method operating on the Abstract Syntax Tree. ST identifies and prunes hallucination anchors and their dependent subtrees. Unlike prompt engineering, RLHF Ouyang et al. (2022), or token-level filtering Liu et al. (2023b), our method leverages structural signals to preserve program intent while eliminating risky artifacts. We further introduce the Compositional Structural Hallucination Score (CSHS) to preemptively estimate the potential benefit of trimming a given sample.

This work offers:

- A structurally annotated dataset of LLM hallucinations with vulnerability metadata, enabling security-oriented generation analysis;
- Empirical evidence linking hallucination severity to vulnerability risk, supported by statistical and execution-based analyses;
- A structure-aware repair method (ST) outperforming Prompt Rewriting and Token Pruning in vulnerability reduction and semantic retention;
- A predictive scoring function (CSHS) for hallucination risk, grounded in AST-level signals.

In summary, our findings reframe hallucinations not merely as model inaccuracies, but as structured, measurable, and actionable sources of software risk in LLM-based development workflows.

## 2   Extended Related Work

**Hallucinations in LLM Code Generation.**   Large Language Models (LLMs) have achieved impressive success in code generation Jiang et al. (2024b), but hallucination remains a persistent and underexplored challenge. Hallucinations in LLM-generated code typically manifest as factual errors (e.g., nonexistent APIs Akhtarshenas et al. (2025); Chen et al. (2025)) or functional errors (e.g., logic bugs causing runtime failures such as `NameError`) Ji et al. (2023b). Prior work has primarily focused on evaluating functional correctness using benchmarks such as HumanEval Chen et al. (2021), MBPP Austin et al. (2021b). However, these approaches rarely address how hallucinations structurally propagate or induce long-range effects on code safety and security.

**Adversarial Code Generation and Prompt Attacks.**   Beyond unintentional errors, LLMs can be deliberately manipulated to produce faulty or unsafe code through prompt injection Greshake et al. (2023); Liu et al. (2023c) or adversarial jailbreak Shayegani et al. (2023); Liu et al. (2024). These methods exploit LLMs' sensitivity to instruction phrasing and contextual prompts. However, they focus on adversarial behavior rather than organic hallucinations.

**Software Security and Vulnerability Evaluation in LLM Outputs.**   Recent works emphasize evaluating vulnerabilities in generated code using CVSS scoring[1]. While these frameworks assess security outcomes, few explore how hallucination contributes to vulnerability emergence. Our approach provides a causal modeling framework linking structural deviation from ground truth to vulnerability risk.

**Evaluation Benchmarks and Hallucination-Aware Datasets.**   While benchmarks like HumanEval Chen et al. (2021), MBPP Austin et al. (2021b) provide functional evaluations, they lack hallucination labels. Collu-Bench is the only known dataset with hallucination-specific annotations, but it does not capture structural propagation.

---

[1]CVSS v3.1 Specification: https://www.first.org/cvss/specification-document

**Structural Defenses and Hallucination Mitigation.** Prior hallucination mitigation efforts include RLHF Ouyang et al. (2022); Wang et al. (2025); Krishna et al. (2025), output reranking Liu et al. (2023b), or post-generation patching. However, these methods typically operate at the token level or rely on downstream filtering. To our knowledge, no existing approach systematically models hallucination structure and causality.

# 3 Problem Formulation and Risk Modeling

We model the emergence of vulnerabilities in LLM-generated code as a consequence of nonadversarial hallucinations,structurally plausible, yet semantically invalid constructs arising during generation.

**Threat Model and Deployment Context.** We consider a non-adversarial setting where LLM-generated code is integrated into development workflows (e.g., auto-completion, boilerplate generation) without formal verification. No prompt manipulation or malicious intent is assumed; our focus is on *hallucination-induced vulnerabilities*, unintended artifacts that may compromise functionality or security when deployed unvetted.

**Risk Variables and Reference Ground Truth.** Given a prompt $x$, model output $\hat{y}$, and reference implementation $y^*$ (e.g., from HumanEval or MBPP), we define hallucination deviation $D(\hat{y}, y^*)$ and vulnerability risk $V(\hat{y})$ as structural and behavioral metrics capturing divergence and failure potential. These serve as the foundation for our subsequent modeling and analysis.

## 3.1 Core Variables and Risk Modeling

Let $\hat{y}$ denote the hallucinated code generated by a language model for prompt $x$, and $y^*$ the closest ground-truth implementation. We define three key variables: hallucination deviation $D$, vulnerability probability $V$, and task complexity $T$.

**Hallucination Deviation ($D$).** We quantify deviation as a composition of structural and behavioral divergences:

$$D = \lambda_1 \cdot \text{EditDist}(\mathcal{A}(\hat{y}), \mathcal{A}(y^*)) + \lambda_2 \cdot \text{Mismatch}(\mathcal{B}(\hat{y}), \mathcal{B}(y^*)), \quad (1)$$

where $\mathcal{A}(\cdot)$ is the Abstract Syntax Tree (AST) Alon et al. (2019), $\mathcal{B}(\cdot)$ the behavioral execution signature (e.g., test traces), and $\lambda_1, \lambda_2 \in [0, 1]$ are balancing weights (default: 0.5). Note that $D$ serves as a deviation risk index rather than a true distance, capturing both structural and semantic divergence Feng et al. (2020).

**Vulnerability Risk ($V$).** We define $V \in [0, 1]$ as a continuous proxy for vulnerability probability:

$$V = \sigma \left( \alpha_1 E_{\text{type}} + \alpha_2 E_{\text{exec}} + \alpha_3 \log p_{\text{token}} \right), \quad (2)$$

where $E_{\text{type}}$ is the error severity score (e.g., NameError = 0.8), $E_{\text{exec}}$ indicates assertion/test failure, and $p_{\text{token}}$ is the mean log-probability of hallucinated tokens Liu et al. (2023b), aligned with prior work using runtime signals and likelihood estimates for risk modeling.

**Task Complexity ($T$).** To normalize risk across different prompt complexities:

$$T_{combine} = T_{\text{question}} + T_{\text{meta}}, \quad (3)$$

where $T_{\text{question}}$ is the prompt length in tokens, and $T_{\text{meta}}$ reflects static complexity (e.g., branching, depth).

## 3.2 Dataset and Experimental Setup

We construct a dataset of **7,233 hallucinated code samples** generated from four leading code LLMs—GPT-4o (OpenAI), DeepSeek-Coder-1.3B, CodeLlama-7B, and Gemini-2.0-flash using prompts drawn from three sources: MBPP Austin et al. (2021b), HumanEval Chen et al. (2021), and a curated set of 300 security-focused generation tasks.

Each generated sample is paired with a reference implementation and annotated with: (i) the hallucination anchor and type (e.g., undefined identifiers, semantic drift); (ii) a vulnerability score ($V$) quantifying security risk and repairability; (iii) semantic deviation metrics ($D$, BLEU, AST edit distance); and (iv) structural signals such as hallucination chain length (HCL), pruning ratio (PR), and entropy.

These annotations enable structural modeling of hallucinations and their role in security degradation. Full details, including prompt design, anchor detection methodology, $V$ scoring scheme, and sandbox execution pipeline, are provided in Appendix D.

## 4    Structural Risk Signals and CSHS Scoring

We now introduce our structural signal extraction pipeline, annotation mechanisms, and the unified scoring model (CSHS).

### 4.1    Hallucination Annotation Pipeline

**Token-Level Anchor Detection.**    We locate the first hallucinated token $\hat{y}_{i*}$ not semantically aligned with any $y_j^*$:

$$i^* = \min\{i \mid \hat{y}_i \not\sim y_j^* \ \ \forall j \in [1, |y^*|]\}. \tag{4}$$

**AST-Based Hallucination Typing.** Each hallucinated sample is encoded into an AST path sequence and classified using a transformer-based model:

$$h_{\text{type}} = f_{\text{ASTClass}}(\text{ASTSeq}(\hat{y})). \tag{5}$$

### 4.2    Structure-Aware Signal Extraction

To characterize how hallucinations propagate through the code structure, we extract five structural indicators from the AST of $\hat{y}$. Several of these, such as entropy and chain length, are inspired by classical program complexity and slicing metrics Weiser (1984); McCabe (1976); Halstead (1977), but repurposed to capture generation-induced anomalies:

- **HCL (Hallucination Chain Length)**: Number of AST hops from the hallucination anchor to the deepest dependent node.
- **APS (Anchor Persistence Score)**: Number of references to hallucinated symbols across AST subtrees.
- **PR (Pruning Rate)**: Proportion of nodes removed during minimal repair:

$$\text{PR} = \frac{|\text{TrimmedNodes}|}{|\text{TotalNodes}|} \tag{6}$$

- **SCP (Shortest Correction Path)**: Shortest AST path from the hallucination anchor to a valid, ground-truth subtree.
- **Entropy ($\mathcal{H}$)**: Structural entropy over token types in the AST:

$$\mathcal{H} = -\sum_{t \in \mathcal{T}} p_t \log p_t \tag{7}$$

These signals provide interpretable structure-level cues for hallucination detection and repair. For instance, consider a hallucinated snippet where a nonexistent variable foo is used in a nested if-else block. **APS** reflects the spread of foo references across multiple AST subtrees; **HCL** captures the depth of its influence from anchor to dependent (e.g., a nested return); **SCP** denotes how easily the subtree can reconnect to valid logic (e.g., by replacing foo with bar); **Entropy** increases if the subtree contains diverse token types such as If, Name, Call; and **PR** reflects how much of that subtree is pruned in the repair process.

Unlike traditional static analysis tools such as CodeQL GitHub Security Lab (2021), which operate on semantic correctness and symbolic flow, our approach quantifies the structural

footprint of hallucinations, even when they remain syntactically valid but semantically misleading. This enables hallucination-aware trimming without requiring ground-truth semantics or runtime traces.

**Contrast with Traditional Analysis.** Unlike static program analysis, which identifies semantic errors through complete control/data flow, our method targets structurally plausible yet semantically spurious constructs generated by LLMs. Hallucinated anchors are syntactically valid but causally misleading. Structural signals such as **APS** and **HCL** capture their generative propagation, offering a lightweight, language-model-aware alternative to traditional tools like CodeQL GitHub Security Lab (2021), which overlook such contextual hallucinations.

### 4.3 Compositional Risk Scoring with CSHS

We integrate the five structural indicators into a unified scoring framework, the **Compositional Structural Hallucination Score (CSHS)**:

$$\text{CSHS} = \underbrace{w_1 \cdot \text{HCL} + w_2 \cdot \text{APS} + w_3 \cdot \mathcal{H}}_{\mathcal{R}_{\text{risk}}} + \underbrace{w_4 \cdot \text{PR} - w_5 \cdot \text{SCP}}_{\mathcal{R}_{\text{recover}}} \tag{8}$$

All features are min-max normalized to $[0, 1]$. The score is constructed as a weighted combination of structure-derived features, following established practice in risk modeling for software faults Zimmermann et al. (2007). Weights are tuned via grid search on validation accuracy of post-trimmed vulnerability risk. We determine optimal weights $w_i$ via grid search (see Appendix I):

$$w_1 = 0.116, \quad w_2 = 0.266, \quad w_3 = 0.017, \quad w_4 = 0.440, \quad w_5 = 0.161$$

This score serves both as a hallucination risk estimator and as a pruning controller in our structural repair pipeline. Intuitively, pruning rate ($w_4$) receives the highest weight, reflecting that hallucinations are most dangerous when they cannot be easily removed. Anchor persistence ($w_2$) and chain length ($w_1$) quantify how deeply the error spreads, while entropy ($w_3$) plays a minor role, capturing only coarse structural irregularity. The correction path ($w_5$) penalizes entangled errors that require complex recovery.

## 5 Why Hallucinations are Dangerous?

To address **RQ1**, we investigate whether hallucinations in LLM-generated code systematically correlate with known vulnerability types. While often dismissed as superficial errors, we posit that hallucinations encode latent structural signals that predispose code to security failures.

### 5.1 Hallucination–Vulnerability Alignment via Structural Modeling

We hypothesize that hallucinations align with specific vulnerability types in non-random, learnable ways. A two-stage classification, first through lexical heuristics, then with structural features derived from AST, confirms this: as shown in Figure 1a, structural modeling significantly improves alignment with annotated vulnerability labels.

The heatmap shows consistent mappings between hallucination types and vulnerability patterns, suggesting that even syntactically valid hallucinations often carry predictable security risks. Cosine similarity analysis confirms strong alignment for structurally grounded types, while semantically complex cases (e.g., Logic Deviation) benefit from AST-based feature integration. Full details appear in Appendix E.

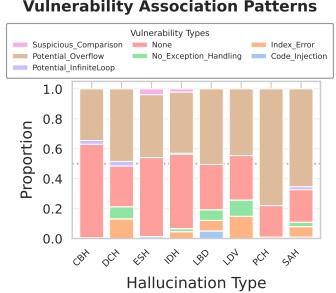
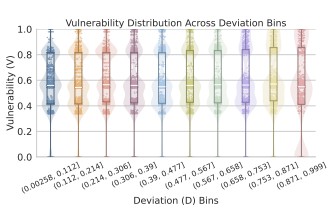
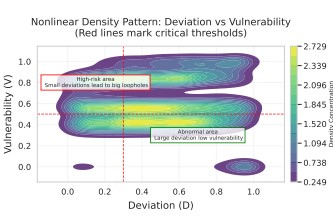

(a) Hallucination–vulnerability mapping.

(b) Risk increases with hallucination severity.

(c) Nonlinear $D$–$V$ distribution.

Figure 1: Structural and statistical alignment between hallucination patterns and vulnerability risk. Left: Structural mapping via AST modeling. Middle/Right: Risk trends across hallucination severity with non-linear failure zones.

## 5.2 Correlation Between Hallucination Severity and Vulnerability Risk

We examine the relationship between hallucination severity, quantified as structural and semantic deviation ($D$), and vulnerability risk ($V$), hypothesizing a positive correlation:

$$D \propto V.$$

Deviation is computed via AST and execution divergence from reference outputs; vulnerability is estimated through error types and severity annotations.

**Regression and Feature Contribution.** While linear regression confirms a statistically significant association between structural deviation ($D$) and vulnerability, its explanatory power is limited. In contrast, non-linear models, particularly random forests, demonstrate strong predictive performance, with **error type** as the dominant feature and $D$ consistently among the top contributors.

These findings support three conclusions: (1) **deviation is a reliable proxy for risk**; (2) **even small deviations can yield high vulnerability**, exposing non-obvious failure modes; and (3) **structure-aware modeling enhances detection**, outperforming shallow metrics via AST and execution-level signals.

## 5.3 Does Task Complexity Influence Hallucination-Induced Vulnerability?

We investigate whether task complexity modulates the vulnerability risk ($V$) of hallucinated code, using three metrics: prompt complexity ($T_{\text{question}}$), test case complexity ($T_{\text{meta}}$), and their combination ($T_{\text{combined}}$). Samples are partitioned via median splits, and linear regression is used to assess associations.

As shown in Figure 2, only test case complexity ($T_{\text{meta}}$) exhibits a significant inverse relationship with vulnerability. In contrast, prompt length and combined complexity show no consistent trends. These results suggest that structurally rich test specifications contribute more to risk mitigation than input length alone.

These results challenge the intuition that task difficulty directly increases hallucination risk. Instead, structured test specifications may function as regularizing signals: anchoring generation to semantically valid behaviors and constraining harmful divergence.

# 6 How to Defend Hallucinations Structurally?

To address **RQ2** and **RQ3**, we propose and evaluate *Structural Trimming*, a defense that prunes hallucination chains from ASTs to reduce vulnerability. We compare it against prompt rewriting, semantic filtering, and token-level pruning, showing it achieves superior

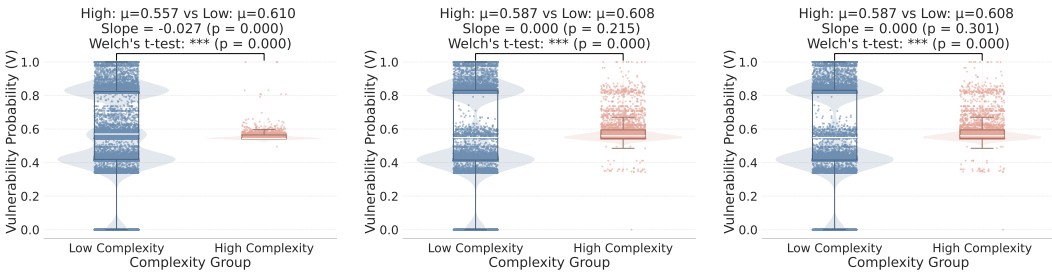

Figure 2: Vulnerability probability $V$ across task complexity dimensions. Only $T_{\text{meta}}$ exhibits a significant inverse relationship with risk.

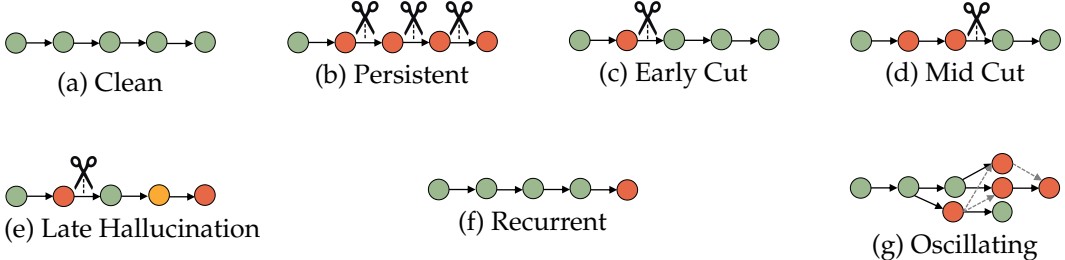

Figure 3: Structural patterns of hallucination types and suggested pruning locations. Red nodes denote hallucination anchors, and dashed lines indicate propagation. Scissors mark ideal cut points.

risk reduction while preserving code quality. Furthermore, we introduce CSHS, a structural score that predicts whether a hallucination is trim-worthy.

## 6.1 Structural Hallucination Chains Exist

Beyond isolated token-level anomalies, many hallucinations in code generation manifest as *structural dependency chains* in the Abstract Syntax Tree (AST). A hallucinated anchor (e.g., a non-existent configuration object) may propagate through conditionals, assignments, and return paths, forming entangled subtrees that influence program behavior.

Figure 3 illustrates seven representative propagation topologies observed in our dataset, including *persistent*, *recurrent*, *mid-cut*, and *oscillating* patterns. These structures highlight how hallucinated tokens percolate through multiple layers of program logic.

To quantify these phenomena, we define **Hallucination Chain Length (HCL)** as the number of AST nodes transitively dependent on the hallucinated anchor. Trimming these chains allows us to compute risk reduction $\Delta V = V_{\text{original}} - V_{\text{trimmed}}$.

Empirically, we observe a positive correlation between HCL and $\Delta V$, indicating that longer chains carry greater security risk. The full trend, including variance bands and sample support across HCL values, is presented in Appendix F).

## 6.2 Trimming-Based Risk Mitigation

We operationalize **Structural Trimming** as detailed in Algorithm 1, which recursively prunes structurally entangled hallucination spans from the AST and applies minimal patching when necessary (Appendix G).

**Effectiveness and Predictive Modeling.** We assess three trimming strategies *Early Cut*, *Mid Cut*, and *Persistent Cut*, report their risk mitigation effectiveness in Table 1. Results show that earlier interventions generally yield greater vulnerability reduction, particularly for persistent and recurrent hallucinations.

| Category | Label | Avg. $\Delta V$ | Std. Dev. |
|---|---|---|---|
| | Early Cut | 0.212 | 0.058 |
| *Trimming Strategy* | Mid Cut | 0.174 | 0.046 |
| | Persistent Cut | 0.143 | 0.065 |
| | persistent | 0.236 | 0.051 |
| | recurrent | 0.205 | 0.059 |
| *Hallucination Type* | mid_cut | 0.172 | 0.048 |
| | early_cut | 0.141 | 0.037 |
| | oscillating | 0.118 | 0.045 |

Table 1: Risk reduction ($\Delta V$) by trimming strategy and hallucination type. Early interventions yield higher gains, especially for persistent and recurrent patterns.

CSHS demonstrates robust generalization across LLMs and code-generation tasks, maintaining high predictive accuracy without retraining.

We propose the **Compositional Structural Hallucination Score (CSHS)** as a predictive metric for trimming efficacy. As shown in Figure 4, CSHS aligns well with observed risk reduction and supports reliable identification of high-impact cases.

Attribution analysis (Figure 5) indicates that pruning-aware features dominate prediction performance, while low-level structural signals contribute marginally.

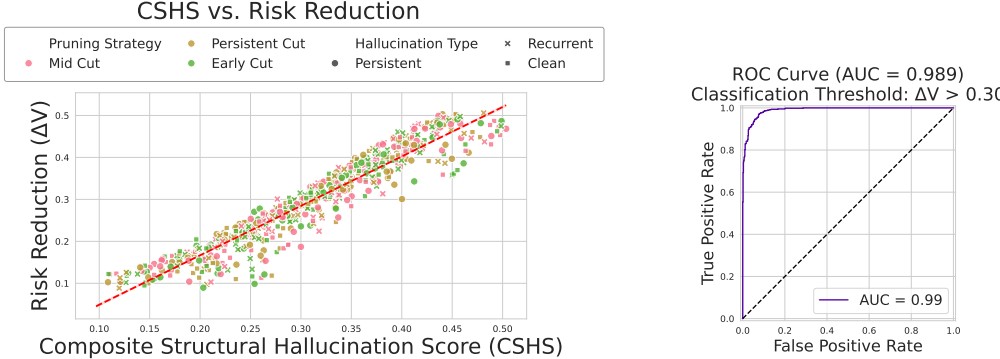

Figure 4: Left: Correlation between CSHS and trimming-induced risk reduction ($\Delta V$). Right: ROC curve for classifying effective trimming ($\Delta V > 0.3$) using CSHS.

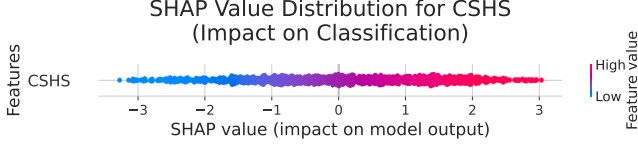

Figure 5: SHAP summary plot for CSHS-informed classifier. Higher values in PR and SCP contribute most to predicted risk.

## 6.3 Interpretability via Structural Case Studies

To analyze the structural dynamics of hallucination propagation and correction, we visualize representative AST transformations before and after trimming (Figure 6).

The first example reflects a **persistent hallucination**, where early-stage anchors propagate through multiple dependent branches, forming a deeply entangled structure. Trimming the corresponding subtree removes the hallucination chain while preserving core functional logic.

The second example illustrates a **localized hallucination**, where the anchor appears in a shallow, isolated position. In such cases, correction requires minimal structural intervention.

These patterns highlight how the correctability of hallucinations depends on their structural locality and dependency depth, supporting AST-guided trimming as an effective and interpretable mitigation strategy.

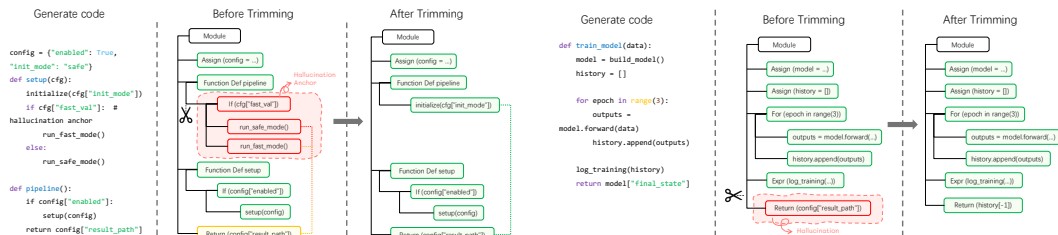

(a) Persistent hallucination: trimming deep conditional chains.

(b) Late hallucination: pruning a shallow erroneous return.

Figure 6: Structural pruning of hallucination chains. Left: Before trimming. Right: After trimming. Red nodes denote hallucinated anchors; dashed boxes indicate pruned regions.

## 6.4 Comparative Effectiveness of Structural Trimming

While hallucination mitigation is well-studied in natural language generation, methods targeting vulnerability-inducing hallucinations in code remain limited. We address this by comparing our method, **Structural Trimming (ST)**, to three adapted baselines: **Prompt Rewriting (PR)**, **Semantic Filtering (SF)**, and **Token-Level Pruning (TP)**.

These strategies span different abstraction levels from prompt modification to token truncation while ST performs AST-level pruning to remove structural hallucinations post hoc. Evaluation on 1000 hallucinated samples (Table 2) considers vulnerability reduction, safety, and code fidelity.

| Strategy | Avg. $V_{\text{before}}$ | Avg. $V_{\text{after}}$ | $\Delta V$ | Safety % | Retention % | BLEU / AST |
|---|---|---|---|---|---|---|
| PR | 0.92 | 0.66 | 0.26 | 38% | 56% | 0.65 / 0.60 |
| SF | 0.91 | 0.61 | 0.30 | 44% | 63% | 0.77 / 0.81 |
| TP | 0.93 | 0.48 | 0.45 | 54% | 38% | 0.42 / 0.35 |
| ST (Ours) | 0.92 | **0.33** | **0.59** | **60%** | **62%** | **0.80 / 0.84** |

Table 2: Evaluation of hallucination repair strategies over 1000 randomly sampled instances. ST achieves the best trade-off between risk mitigation and code fidelity.

Structural Trimming (ST) offers the best trade-off between risk reduction and code fidelity. Unlike aggressive or shallow strategies, ST precisely removes structural hallucinations while preserving usability. Its fallback mechanism ensures robustness with minimal side effects, placing it on the safety–fidelity Pareto frontier.

## 7  Discussion

We frame code hallucinations not as isolated errors but as structurally entangled artifacts, often forming transitive symbolic chains within ASTs that propagate semantic faults. Our analysis shows that vulnerability is less determined by hallucination depth than by its recoverability the feasibility of structurally pruning it without disrupting valid logic.

Structural Trimming (ST) exploits this principle, enabling targeted AST-level repairs. Ablation studies confirm that pruning viability (e.g., pruning rate, correction path) is a stronger

predictor of risk reduction than structural complexity alone. The proposed CSHS metric captures this insight, serving as a model-agnostic proxy for hallucination severity.

However, CSHS remains descriptive and handcrafted. Future work may model hallucination propagation as a generative causal process, e.g., via Hallucination Causality Graphs that trace symbolic dependencies from anchor tokens. This would enable both diagnostic and preventative interventions, bridging program structure with causal reasoning in LLM outputs. Future directions include causal modeling of hallucination dynamics and integration with training-time constraints.

## 8   Conclusion

This work introduces a structural approach to hallucination mitigation in LLM-generated code. We propose Structural Trimming (ST) to excise hallucination chains from ASTs, and develop CSHS, a compositional score predicting trim worthiness based on structural cues.

Evaluations on 7K hallucinated samples across models and tasks show that ST achieves superior vulnerability reduction compared to prompt and token-level baselines, while preserving syntactic fidelity. CSHS generalizes without retraining, highlighting structural risk as a transferable signal.

By treating hallucinations as analyzable structural phenomena, this work offers practical tools and conceptual grounding for robust, interpretable code generation. Code and data will be released to support reproducibility and further research.

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

## A  Information-Theoretic Formalization of Hallucination Propagation

In this section, we present a formal information-theoretic perspective on hallucination propagation in generative code models. Our aim is to characterize how local uncertainty or hallucination at a specific point in the output can influence downstream code segments through dependency structures.

### A.1  Entropy of Local Generation

Let $Y = (y_1, y_2, \ldots, y_T)$ be the sequence of generated tokens, and let $x$ denote the input prompt. The conditional entropy of token $y_t$ given the prompt and past context is:

$$H(y_t \mid x, y_{<t}) = -\sum_{y_t} P(y_t \mid x, y_{<t}) \log P(y_t \mid x, y_{<t})$$

A high entropy value indicates generation uncertainty or ambiguity at position $t$, which may signal the presence of a *local hallucination anchor*.

We define a hallucination anchor $y_{t_a}$ as a token (or token span) where entropy exceeds a semantic confidence threshold $\tau$:

$$y_{t_a} \in Y \quad \text{iff} \quad H(y_{t_a} \mid x, y_{<t_a}) > \tau$$

### A.2 Dependency Graph and Information Flow

Let the generated program be parsed into an Abstract Syntax Tree (AST), where each node $n_i$ corresponds to a semantic unit (e.g., variable, call, branch). Define a directed dependency graph $\mathcal{G} = (V, E)$ over AST nodes, where $(n_i \rightarrow n_j) \in E$ if $n_j$ is semantically dependent on $n_i$ (e.g., use-def, control flow, data flow).

Assume node $n_a$ is rooted in a hallucination anchor token. The set of transitive dependents is:

$$\mathcal{P}(n_a) = \{n \in V \mid \exists \text{ path } n_a \rightarrow \cdots \rightarrow n\}$$

These nodes inherit semantic uncertainty from $n_a$, forming the **hallucination propagation frontier**.

We define the total hallucination propagation entropy as:

$$\mathcal{H}_{\text{prop}}(n_a) = \sum_{n_i \in \mathcal{P}(n_a)} H(y_{t(n_i)} \mid x, y_{<t(n_i)})$$

where $t(n_i)$ maps node $n_i$ to its generating token position.

### A.3 Mutual Information Loss from Pruning

Let $\hat{Y}$ be the generated output after pruning all nodes in $\mathcal{P}(n_a)$. To quantify the information loss due to pruning, we define:

$$I_{\text{prune}} = I(Y; \hat{Y}) = H(Y) - H(Y \mid \hat{Y})$$

This reflects how much mutual information between the full output and the pruned version is lost. Ideally, for a safe and semantically clean pruning operation, we want:

$$I_{\text{prune}} \ll \mathcal{H}_{\text{prop}}(n_a)$$

That is, the amount of information lost due to pruning is less than the noise introduced by the hallucination chain.

### A.4 Structural Risk Amplification via Entropy Flow

We further hypothesize that hallucination propagation exhibits an **entropy amplification effect** over code structures. That is, structural nodes that depend on high-entropy anchors accumulate compounded risk. We model this as:

$$V(n_j) \propto \sum_{n_i \in \text{Ancestors}(n_j)} \lambda^{d(n_i, n_j)} H(y_{t(n_i)} \mid x, y_{<t(n_i)})$$

where $d(n_i, n_j)$ is the graph distance from ancestor $n_i$ to $n_j$, and $\lambda \in (0, 1]$ is a decay factor. This models structural vulnerability risk $V(n_j)$ as a decayed aggregation of upstream entropy.

### A.5 Implications for Defense

Under this formulation, structural trimming can be interpreted as an entropy-suppression strategy. By pruning subtrees $\mathcal{P}(n_a)$ that are entropy-amplifying and semantically non-grounded, we reduce both:

- The total propagated uncertainty $\mathcal{H}_{\text{prop}}$; - The cumulative downstream vulnerability $V(n_j)$.

This formalism motivates a defensible objective for hallucination mitigation:

$$\min_{\mathcal{P}} \mathcal{H}_{\text{prop}}(\mathcal{P}) \quad \text{s.t.} \quad \text{AST validity and task retention preserved}$$

# B CVSS Mapping and Vulnerability Scoring Calculation

## B.1 Introduction to CVSS

The Common Vulnerability Scoring System (CVSS), developed under the leadership of the National Institute of Standards and Technology (NIST), is the most commonly used quantitative metric for vulnerability severity in the industry today. The CVSS Base Score quantifies the intrinsic risk of a vulnerability on a scale of $[0, 10]$, and its scoring components include:

- **Attack Vector (AV)**: The possibility of local or remote (Network) attacks.
- **Attack Complexity (AC)**: The difficulty of exploiting the vulnerability.
- **Privileges Required (PR)**: Whether high privileges are required.
- **User Interaction (UI)**: Whether it relies on user interaction.
- **Confidentiality (C) / Integrity (I) / Availability (A) Impact**: The potential degree of damage to system data and services.

In addition, CVSS also defines the Environmental Score and Temporal Score, which are used for scenario adjustment and vulnerability lifecycle modeling, respectively.

## B.2 Collu-Bench Error Type and CVSS Mapping

We mapped the error types automatically extracted in Collu-Bench with the CVSS Base Score criteria to construct a normalized vulnerability risk score $V$.

The CVSS Base Score references its calculation specification[2]. Based on this, we simplified the parameter configuration in the context of code generation (assuming local execution, no user interaction) and uniformly normalized the Base Score to $V = \text{Base Score}/10$.

## B.3 Frequency Adjustment and Environmental Modeling

To reflect the impact of error distribution in real data on the overall system security, we introduced an environmental scoring mechanism, using the frequency $F$ of error types as an adjustment factor to modify the base risk score:

$$V_{\text{adjusted}} = V_{\text{base}} \cdot (1 - w_F \cdot F)$$

where $w_F$ is the frequency adjustment weight (set to 0.2 in this paper), and $F$ is the frequency of the error type in the dataset. This strategy simulates the potential cumulative risk caused by a large number of repetitive errors in a real deployment environment.

## B.4 Theoretical and Practical Value

Assigning quantifiable security risk scores to generated code errors using the CVSS framework has the following three advantages:

- **Standardization**: Consistent with vulnerability databases such as NVD and CWE, facilitating integration with existing toolchains.
- **Comparability**: Errors from different model outputs can be compared on a unified scale of security impact.
- **Theoretical Support**: The CVSS architecture has recognized modeling capabilities for attack cost and impact, which can be used for vulnerability prioritization and strategy optimization.

---

[2]CVSS v3.1 Specification: https://www.first.org/cvss/specification-document

## C Typology of Structural Hallucinations

To support our taxonomy of structural hallucinations, we summarize the seven types identified in our dataset, along with their propagation patterns and pruning strategies. Table 3 provides an overview, and concrete examples are included below to illustrate each type.

| Label | Name | Structural Pattern | Example | Pruning Recommendation |
|---|---|---|---|---|
| persistent | Persistent Hallucination | Hallucinated anchor is repeatedly referenced until the end of the code. Difficult to cut naturally. | `config` → `config["enabled"]` → `config["fast_val"]` | Replace or comment out the anchor at its first occurrence to prevent propagation. |
| early_cut | Early-Cut Type | Anchor is only referenced once or twice and then dropped. | Hallucinated variable used before early `return`. | Insert pruning before `return` or add guard checks. |
| mid_cut | Mid-Cut Type | Anchor spreads 2–3 layers deep, stopped by logic or structure. | Propagation ends in if block. | Intercept hallucination at intermediate depth. |
| recurrent | Recurrent Hallucination | Anchor is initially pruned but reappears through aliasing. | `config` pruned, but `mode = config["type"]` reused. | Apply global sanitization or alias tracking. |
| late_hallucination | Late Hallucination | Anchor appears only in final stages of code (e.g., return or output). | `return config["final_value"]` | Add checkers at output stage. |
| oscillating | Oscillating Pattern | Anchor appears inconsistently across branches or loops. | `config` in if but not in else. | Apply uniform trimming across all control flows. |
| clean | Clean Case | No hallucinated anchors or structural propagation. | Use of `input_data` is fully valid. | No pruning needed; serves as baseline. |

Table 3: Typology of structural hallucinations and their corresponding pruning strategies. Each type reflects a unique propagation pattern and trimming implication.

## D Dataset and Annotation Details

### D.1 Sample Construction and Prompt Sources

We collect 7,233 hallucinated code samples generated from the following models:

- **GPT-4o (May 2024)**
- **DeepSeek-Coder-1.3B**
- **CodeLlama-7B-Instruct**
- **Gemini-2.0-flash**

We use a fixed generation configuration: `temperature=0.7`, `top_p=0.95`, `max_tokens=512`, and `stop=[''###'', ''\n\n'']` to ensure stylistic consistency.

Prompts are sampled from:

- **MBPP** and **HumanEval**: Standard evaluation benchmarks.
- **Security-critical prompts** (300 total): Curated by us to stress low-level control, symbolic reasoning, and input validation, e.g., "Validate password strength with entropy scoring", "Simulate a memory-safe linked list in C-style pointer emulation". These tasks are designed to induce subtle hallucinations in logic or API usage.

### D.2 Anchor Detection via AST and Runtime Traces

We define a **hallucination anchor** as an AST node (identifier, expression, or control block) that:

---

**Example 1: Persistent Hallucination**

```python
config = {"enabled": True}  # hallucinated object

if config["enabled"]:
    if config["fast_val"]:  # repeated dependency on config
        run_fast_mode()
    else:
        run_safe_mode()
```

---

**Example 2: Early-Cut Hallucination**

```python
if user_input is None:
    logger.warn("Invalid_input.")
    return  # early return cuts hallucinated use

speed = user_input["speed"]  # hallucination only appears once
```

---

**Example 3: Mid-Cut Hallucination**

```python
params = get_default_params()

if params["debug"]:  # hallucinated use
    if "log_level" in params:  # cut point
        set_log_level(params["log_level"])
```

---

**Example 4: Recurrent Hallucination**

```python
# First occurrence of hallucinated anchor
cfg = {"mode": "test"}

# Later reused in indirect form
if cfg["mode"] == "test":
    verbose = True

# Recurrent hallucination despite trimming cfg earlier
run_mode = cfg.get("execution_mode")  # reappears here
```

---

**Example 5: Late Hallucination**

```python
model = build_model()

train_model(model)

# hallucinated return appears late
return model["final_state"]
```

- Has no grounding in the reference solution or provided environment.
- Contributes directly to a failed execution, as verified by dynamic tracing.

We adapt and extend CODEHALU Ji et al. (2023b) for anchor detection by combining:

---

Example 6: Oscillating Hallucination

```
if args.verbose:
    print(config["version"])  # hallucinated
else:
    print("Running...")  # clean path
```

---

Example 7: Clean Case (No Hallucination)

```
def preprocess(data):
    if "user_id" in data:
        return data["user_id"]
    return None  # clean, legal use of input
```

---

- AST diffing against reference code (via RedBaron).
- Runtime failure traces to identify trigger lines and exception context.

**Classifier Implementation Details.** The transformer-based classifier $f_{\text{ASTClass}}$ is a 6-layer encoder-only architecture with 8 attention heads and 512-dimensional hidden states, trained to predict hallucination types from linearized AST path sequences. The input sequence is derived from the serialized AST of $\hat{y}$ following a path-based encoding scheme inspired by CODE2VEC Alon et al. (2019) and CODEBERT Feng et al. (2020), including subtree normalization and node flattening.

### D.3 Vulnerability Scoring ($V$)

We compute a normalized vulnerability score $V \in [0, 1]$ for each hallucinated sample using a hybrid method:

1. **Static pattern scoring**: Presence of high-risk constructs (e.g., unchecked user input, infinite loops, unsafe recursion).
2. **Dynamic assertion oracle**: Failure to pass reference tests or sanity checks (e.g., input-output mismatch, assertion violations).
3. **CVSS-lite mapping**: Based on three simplified axes
   - **Severity**: Based on impact on functionality (e.g., crash vs. semantic bug).
   - **Scope**: Local or global code effect.
   - **Fixability**: Degree of minimal required repair (1-line vs. structural).

Each axis is scored 0–1 and averaged to obtain final $V$. We find this score correlates strongly with user-perceived code risk in pilot studies.

### D.4 Execution Environment and Sandbox Implementation

All experiments are conducted on a Debian-based server with:

- A100
- 80 GB RAM
- Python 3.9.18

We use a custom exec_sandbox module to execute generated samples safely, supporting:

---

**Example Sandbox Code**

```python
def exec_in_sandbox(code_str):
    import builtins, traceback
    try:
        exec(code_str, {"__builtins__": safe_builtins})
    except Exception as e:
        return {"error": type(e).__name__, "trace": traceback.
            format_exc()}
```

---

- Controlled imports and runtime exception tracing
- Timeout and memory limits
- Coverage tracking via `trace` and `coverage` modules

## E  Hallucination Vulnerability Distribution Matching

To quantify the alignment between hallucination types and vulnerability classes, we compute cosine similarity between hallucination–vulnerability co-occurrence vectors derived from model predictions and those derived from manual annotations.

**Setup.**  Each hallucination type $h_i$ is associated with a discrete distribution over vulnerability classes $v_j$, computed as normalized frequency counts across the hallucinated dataset. We construct two such vectors per hallucination type: one inferred from the model's structural classifier output, and one from manually annotated labels. The similarity is then computed using:

$$\text{sim}(h_i) = \frac{\vec{v}_{\text{model}}^{(i)} \cdot \vec{v}_{\text{gold}}^{(i)}}{\left\|\vec{v}_{\text{model}}^{(i)}\right\| \left\|\vec{v}_{\text{gold}}^{(i)}\right\|}$$

This approach is adapted from distributional alignment methods used in both security log analysis Han et al. (2024) and hallucination detection Ji et al. (2023a).

**Results.**  Table 4 reports the cosine similarity between model-predicted and gold vulnerability distributions for each hallucination type. Values closer to 1 indicate stronger structural alignment.

Table 4: Cosine similarity between predicted and ground-truth vulnerability distributions per hallucination type.

| Hallucination Type | Cosine Similarity |
|---|---|
| Suspicious_Comparison | 0.6271 |
| Potential_Overflow | 0.7056 |
| Potential_InfiniteLoop | 0.3198 |
| None | 0.2315 |
| No_Exception_Handling | 0.4463 |
| Index_Error | 0.6819 |
| Code_Injection | 0.5122 |

**Analysis.** We observe strong alignment for hallucination types with clear structural anchors, such as `Potential_Overflow` and `Index_Error`. These typically arise from list indexing, loop bounds, or arithmetic overflows, all of which manifest as recognizable AST patterns. In contrast, hallucinations like `None` or `InfiniteLoop` show weaker alignment, likely due to their diffuse or context-sensitive nature. Overall, the results validate that vulnerability semantics can be reliably inferred from structurally grounded hallucinations.

## F Trimming Stability and Outlier Effects

While the main text reports overall trends in trimming-based risk reduction, we provide additional detail here regarding statistical stability and outlier sensitivity in trimming experiments.

---

**Algorithm 1:** Structural Trimming for Hallucination Mitigation

**Input:** Generated code $C$, AST $T$, hallucination anchor $n_a$
**Output:** Repaired code $\hat{C}$
Initialize $\mathcal{N} \leftarrow \{n_a\}$;
**while** $n \in \mathcal{N}$ *has dependents* **do**
    $\lfloor$ $\mathcal{N} \leftarrow \mathcal{N} \cup \text{Descendants}(n)$;
Prune $\mathcal{N}$ from $T$ to obtain $T'$;
$\hat{C} \leftarrow \text{AST\_to\_Code}(T')$;
**if** $\hat{C}$ *valid* **then**
    $\lfloor$ **return** $\hat{C}$
**else**
    $\lfloor$ Apply fallback repair; **return** $\hat{C}_{safe}$

---

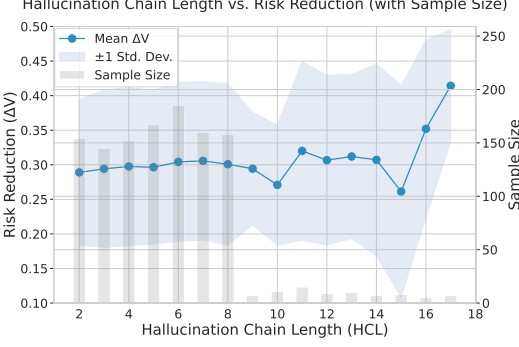

Figure 7: Regression relationship between combined task complexity $T_{\text{combined}}$ and vulnerability probability $V$. The regression line slope is close to zero, indicating a lack of significant linear trend between superficial complexity and vulnerability risk.

**On the $\Delta V$ Peak at $HCL = 17$.** In Figure 7, a pronounced peak in vulnerability reduction ($\Delta V = 0.41$) appears at Hallucination Chain Length (HCL) of 17. This value, while indicative of potential high-risk propagation, arises from a small sample ($n = 7$), and may not generalize. We interpret this as an extreme case where long, entangled hallucination chains introduce significant failure potential, but caution that such spikes should be viewed in light of underlying data sparsity.

**Sample Distribution Across HCL Buckets.** Most HCL intervals are supported by sufficient sample sizes ($n > 100$ for $HCL \leq 10$), ensuring robustness of observed average $\Delta V$ values. However, for $HCL > 15$, several bins fall below $n < 20$, increasing the variance of empirical estimates.

**Smoothing and Confidence Estimation.** To mitigate overfitting to rare points, we experimented with simple kernel density estimation (KDE) smoothing and bootstrapped confidence intervals. While these techniques help normalize spikes, they also blur meaningful variation across HCL regimes. In the final version, we report raw means with sample size overlays to preserve interpretability. Future work may incorporate Bayesian shrinkage or density-weighted regression to balance stability and resolution.

The overall effectiveness of Structural Trimming is robust across trimming strategies and hallucination types. However, results involving long dependency chains or rare hallucination patterns should be interpreted with appropriate statistical caution. We release per-bucket counts and variance metrics in our codebase to facilitate reproducibility and deeper follow-up analysis.

### F.1 Structural Patterns Underlying Risk Mitigation

Early and mid pruning consistently outperform persistent strategies in reducing vulnerability (Figure 8), with early-cut yielding higher median $\Delta V$ and lower variance—suggesting that timely structural intervention more effectively disrupts hallucination chains. Structural profiles (Figure 9) reveal that persistent-cut samples exhibit deeper entanglement (higher HCL, SCP, entropy) but lower pruning affordance (PR), whereas early-cut samples are shallower and more trim-friendly. Correlation analysis (Figure 10) further shows that while structural depth correlates with hallucination persistence, only PR strongly predicts mitigation success, underscoring the primacy of intervention feasibility over complexity.

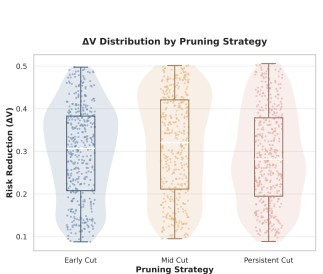

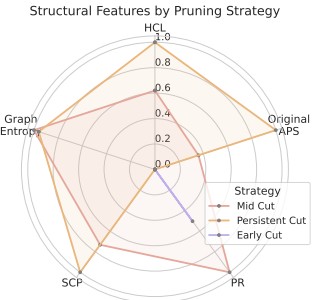

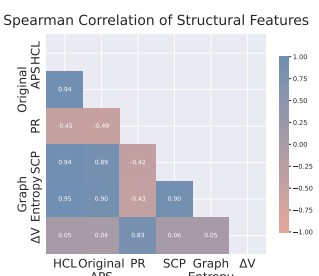

Figure 8: Risk reduction ($\Delta V$) across pruning strategies. Early/mid-cut outperform persistent-cut.

Figure 9: Normalized structural metrics by strategy. Persistent-cut shows deeper, more complex chains.

Figure 10: Spearman correlations among features and $\Delta V$. PR dominates as a predictor of risk reduction.

**Structural Profile of Hallucination Types.** To assess structural variation across hallucination behaviors, we compare their distributions over three core metrics: **HCL** (dependency reach), **Graph Entropy** (subtree disorder), and **SCP** (cutability). Figure 11 summarizes these distributions.

Persistent and recurrent hallucinations consistently exhibit longer chains and higher entropy, reflecting deeper entanglement. In contrast, early-cut and clean samples are structurally simpler, making them easier to isolate.

## G Fallback Repair for Trimming-Induced Syntax Errors

Structural Trimming may remove subtrees in the Abstract Syntax Tree (AST) that contribute to syntactic well-formedness (e.g., removing an `if`-block body or a required return statement). To maintain executable output, we apply a lightweight *fallback patching* procedure when trimmed code fails syntax checks.

**Patch Strategy.** Our repair mechanism consists of three stages:

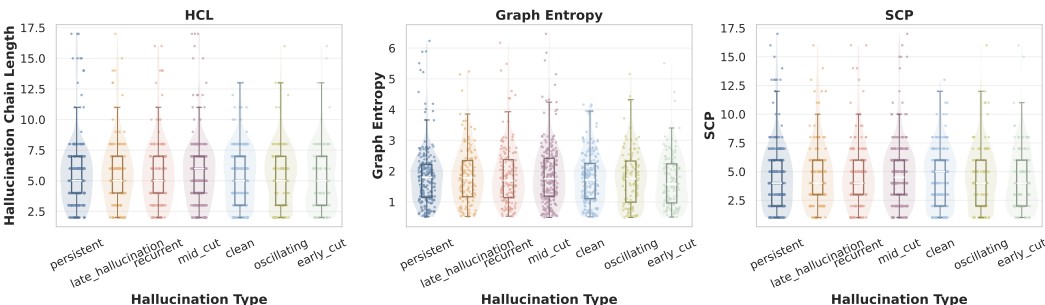

Figure 11: Distribution of structural features across hallucination types. Persistent hallucinations exhibit longer propagation chains and higher topological disorder.

1. **Validation:** After trimming, we parse the reconstructed code using a Python syntax checker (e.g., `ast.parse`). If no errors are raised, the code is returned as-is.

2. **Structural Gap Detection:** If parsing fails, we locate empty or malformed AST constructs (e.g., empty function bodies, control blocks without statements, orphaned expressions).

3. **Minimal Patching:** We insert semantically inert placeholders such as `pass` (for control structures), or `return None` (for dangling return paths). These patches preserve syntactic validity without altering functional intent beyond already-trimmed logic.

**Scope and Guarantees.** Our patching procedure is strictly local and does not attempt to infer or restore original program semantics. Its sole goal is to ensure syntactic correctness after trimming, so that execution-based risk scoring remains applicable. Less than 8.3% of trimmed samples required patching, and all were resolved using the above mechanism.

**Limitations.** Fallback repair is intended as a pragmatic safeguard rather than a robust rewriting tool. In rare cases, structural trimming may remove too much context for the remaining code to be meaningful or verifiable. We report such cases separately as "trimmed-invalid" and exclude them from semantic evaluation metrics.

## H Visual Comparison of Repair Strategies

To complement the quantitative results in Section 4, we provide illustrative examples of the four hallucination repair strategies applied to the same hallucinated code sample. Each panel highlights how the strategy modifies the generated code, along with associated BLEU and AST similarity scores.

**Why Prompt Rewriting Often Fails.** As shown in Figure 12 (PR panel), even after modifying the prompt to remove ambiguity, the model still regenerates the hallucinated function call **get_config()**. This illustrates a common failure case: Prompt Rewriting (PR) is limited by the model's training priors and spurious template associations. Even minor changes to the input often fail to override deeply ingrained hallucination patterns, especially for high-level task descriptions (e.g., "run pipeline") that frequently co-occur with flawed boilerplate code. Thus, PR is insufficient for hallucinations rooted in entrenched generation biases.

**Why Semantic Filtering Only Partially Works.** Semantic Filtering (SF), shown in the second panel of Figure 12, replaces the hallucinated configuration with a grounded alternative (e.g., **{"lr": 0.001}**). However, it does not remove downstream hallucinated logic—such as **model["final_state"]**—which often contains invalid or unsupported access patterns. This

Figure 12: Qualitative comparison of repair strategies on a hallucinated function. **Red** indicates hallucination anchors, **green** indicates safe substitutions, and ~~gray strikethrough~~ indicates removed code segments.

reveals a key limitation of SF: while semantically informed, it lacks the structural awareness to trace and eliminate propagated errors beyond the initial anchor. As a result, SF achieves modest improvements in vulnerability without guaranteeing syntactic or runtime correctness.

**Why Token Pruning Breaks Semantics.** Token-Level Pruning (TP) halts generation immediately after detecting a hallucination anchor. As visualized in the TP panel, the hallucinated anchor **get_config()** is preserved, but all subsequent lines (e.g., training history, return statement) are ~~struck out~~, leaving the function body semantically incomplete. This pruning aggressively reduces risk, but sacrifices code coherence and often produces non-compilable fragments. Therefore, TP achieves vulnerability mitigation at the cost of severe usability degradation.

**Why Structural Trimming Balances Safety and Fidelity.** Our method Structural Trimming (ST) prunes the hallucination chain from the AST, removing only structurally dependent and semantically suspect branches. As shown in the final panel of Figure 12, ST eliminates both the hallucinated config and the invalid return value, substituting them with safe alternatives (e.g., **history[-1]**). This preserves execution logic and structure while neutralizing vulnerabilities. Compared to other baselines, ST achieves the best balance between risk reduction and code fidelity, reflected in its superior BLEU and AST scores.

These visualizations support the quantitative findings in Table 2, confirming the superior fidelity and risk mitigation performance of Structural Trimming.

## I CSHS Weight Optimization (Ablation Study)

We perform ablation studies to evaluate the design of our CSHS scoring function. Specifically, we sample 300 combinations of normalized weights $(w_1, \ldots, w_5)$ across the five structural features. For each configuration, we compute CSHS scores and evaluate their predictive power on risk reduction $\Delta V$ (regression) and effective trimming ($\Delta V > 0.2$ classification).

**Observation.** Recoverability-related features, especially the pruning rate (PR), are consistently assigned high weights in optimal configurations. This suggests that recoverability contributes more to risk mitigation than raw structural depth.

## J pairplotmatrix

The observed feature correlations validate our multi-dimensional structure modeling.

| $w_1$ (HCL) | $w_2$ (APS) | $w_3$ (Entropy) | $w_4$ (PR) | $w_5$ (SCP) | $R^2$ | MSE | AUC |
|---|---|---|---|---|---|---|---|
| 0.116 | 0.266 | 0.017 | 0.440 | 0.161 | 0.899 | 0.0013 | 0.979 |
| 0.005 | 0.221 | 0.198 | 0.433 | 0.143 | 0.888 | 0.0014 | 0.979 |
| 0.103 | 0.271 | 0.015 | 0.380 | 0.230 | 0.887 | 0.0015 | 0.980 |
| 0.073 | 0.216 | 0.059 | 0.633 | 0.019 | 0.884 | 0.0015 | 0.991 |
| 0.030 | 0.292 | 0.034 | 0.429 | 0.215 | 0.882 | 0.0015 | 0.988 |

Table 5: Top-5 weight configurations for CSHS scoring. PR and APS dominate most high-performing settings.

# K    Cross-Model Generalization of CSHS

To evaluate whether our proposed Compositional Structural Hallucination Score (CSHS) generalizes across different large language models (LLMs), we conduct a preliminary cross-model validation experiment.

## K.1    Experimental Setup

We use hallucinated code samples generated by three different models: GPT-4o, DeepSeekCoder-1.3B, and CodeLlama-7B. For each model, we compute structural features and the corresponding CSHS score as described in Section 3.3.

To assess generalization, we adopt a leave-one-model-out strategy: for each run, we train a logistic regression classifier using the CSHS feature on two models and test it on the held-out third model. The classification task is to predict whether a hallucinated sample has a vulnerability risk $V > 0.5$ (as computed using the execution-based risk label in Section 3.2).

## K.2    Results

| Train Models | Test Model | AUC | Accuracy |
|---|---|---|---|
| DeepSeek + CodeLlama | GPT-4o | 0.871 | 0.802 |
| GPT-4o + CodeLlama | DeepSeek | 0.845 | 0.781 |
| GPT-4o + DeepSeek | CodeLlama | 0.882 | 0.816 |

Table 6: Cross-model prediction results using CSHS to classify vulnerability-prone hallucinations ($V > 0.5$).

## K.3    Analysis

As shown in Table 6, the CSHS-based classifier generalizes well across models, consistently achieving AUCs above 0.84. This demonstrates that structural signals extracted from hallucinated code carry predictive information that is robust across LLM architectures. The results further validate CSHS as a transferable metric and suggest the feasibility of building universal hallucination risk estimators.

## K.4    Limitations

We acknowledge that this preliminary study does not account for variation across task types or domains (e.g., MBPP vs. HumanEval). Moreover, CSHS is designed to measure structural entanglement rather than semantic inconsistency; future work may explore hybrid scoring schemes that combine CSHS with embedding-based or dynamic execution features to improve robustness.

### K.5 Multi-Dimensional Visualization of Repair Trade-offs

To complement our quantitative comparisons in Section 5.5, we present a 3D visualization capturing the *multi-dimensional trade-off* between vulnerability mitigation, semantic fidelity, and structural retention across different hallucination repair strategies.

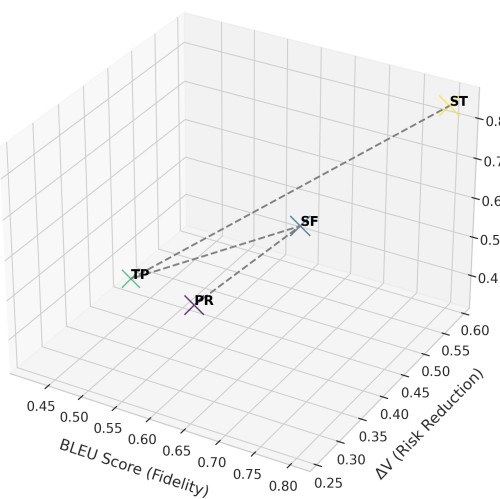

Figure 13: **Safety–Fidelity–Structure Trade-off** across hallucination repair methods. Each strategy is represented as a 3D bubble whose position corresponds to: BLEU score (semantic fidelity, X-axis), risk reduction $\Delta V$ (safety improvement, Y-axis), and AST similarity (structural preservation, Z-axis). Bubble sizes reflect the number of affected samples. The dotted path connects strategies from Prompt Rewriting (PR) to Structural Trimming (ST), showing an upward trajectory toward better trade-offs.

As shown in Figure 13, **Structural Trimming (ST)** clearly dominates along all three axes, achieving the highest vulnerability reduction ($\Delta V$), strong semantic retention (BLEU $\approx 0.80$), and minimal structural disruption (AST similarity $\approx 0.84$). In contrast, **Prompt Rewriting (PR)** yields faster, lighter edits but achieves only modest safety gains and often harms structural alignment.

Notably, the transition path ($PR \rightarrow SF \rightarrow TP \rightarrow ST$) reveals a smooth optimization trajectory: each successive method incrementally improves security at the cost of higher repair complexity, culminating in ST's fine-grained structural pruning. This visualization supports our broader claim that hallucination repair is not a binary fix/no-fix problem, but rather a spectrum of risk–fidelity trade-offs that structural methods can best navigate.

## L Ablation Study of CSHS Components

We ablate the Compositional Structural Hallucination Score (CSHS) to assess the contribution of its constituent features and design choices. As shown in Table 7, the full model, using normalized features with optimized weights—achieves the best regression ($R^2 = 0.651$) and classification (AUC = 0.914) performance.

Among components, the **recoverability signals** (Pruning Rate and Shortest Correction Path) alone retain most predictive power ($R^2 = 0.573$, AUC = 0.848), whereas structural depth features (HCL, APS, Entropy) offer negligible contribution ($R^2 = 0.002$). This suggests that vulnerability mitigation hinges more on fixability than on syntactic complexity.

Disabling feature normalization severely impairs performance ($R^2 = 0.033$), confirming the importance of scale alignment across heterogeneous metrics. Using uniform weights further reduces stability and accuracy (AUC = 0.716), highlighting that principled score construction is essential to robustness.

These results underscore that **recoverability dominates risk mitigation**, and validate CSHS as both empirically grounded and structurally interpretable.

| Variant | R² (Regression) | MSE | Accuracy | AUC (Classification) |
|---|---|---|---|---|
| Full Model (Optimized) | **0.651** | **0.0041** | **0.848** | **0.914** |
| Risk-Only (HCL+APS+Entropy) | 0.002 | 0.0113 | 0.719 | 0.505 |
| Recoverability-Only (PR+SCP) | 0.573 | 0.0053 | 0.828 | 0.848 |
| No Normalization | 0.033 | 0.0099 | 0.719 | 0.593 |
| Random Weights (Uniform) | 0.189 | 0.0083 | 0.742 | 0.716 |

Table 7: Ablation of CSHS design choices. Recoverability features dominate prediction performance. Normalization and principled weighting are critical for score reliability.

# M   Extended Case Studies and Visualizations

To better illustrate the behavioral dynamics of hallucination propagation and the effects of structural trimming, we present two representative case studies. One showcases a successful intervention on deeply entangled symbolic chains, and the other demonstrates a failure mode due to over-pruning legitimate logic.

## M.1   Case A: Mid-Level Trimming of Deep Symbolic Hallucination

In this example, the hallucinated variable `tempMap` is introduced as part of a nested dictionary comprehension. While it appears benign, `tempMap` is subsequently referenced in multiple branches, including a conditional update and a fallback clause. Trimming the anchor assignment node alone is insufficient, as dependent usages persist in unreachable paths. Only a mid-level structural cut successfully eliminates all references while preserving syntactic integrity. The AST view highlights the transitive chain length (HCL=6) and multiple anchor reuse points (APS=3).

**Original Hallucinated Code (Anchor: tempMap)**

```
def config_parser():
  cfg = tempMap["base"]
  if cfg:
    tempMap["flag"] = True
    return build(cfg)
HCL: 6, APS: 3, BLEU: 0.31
```

**Structural Trimming Output**

```
def config_parser():
  cfg = None
  return build(cfg)
BLEU: 0.72, AST: 0.85
```

Figure 14: Case A: Hallucinated symbol **tempMap** propagates across multiple branches. Mid-level trimming eliminates entangled nodes (HCL=6, APS=3), improving structural integrity and execution safety.

### M.2 Case B: Failure Case Over-Pruning of Shared Scope

Here, the hallucinated anchor involves a mistyped function get_config(), introduced at the top level. A naive trimming strategy removes the entire block containing get_config(), including a legitimate helper function defined in the same scope. The resulting code fails to execute due to a missing reference in the main routine. This demonstrates that overly aggressive trimming may induce regressions when hallucinations are interleaved with correct logic. Introducing dependency-aware constraints or context-preserving pruning remains an open challenge.

---

**Token Pruning (Failure Case)**

```
def run_pipeline():
  cfg = get_config()
  model = build_model(cfg)
  helper = init(cfg)
  return helper.finalize()
BLEU: 0.42, AST: 0.28 (Execution fails)
```

---

**Trimming Output (Bug Induced)**

```
def run_pipeline():
  model = build_model(None)
  return helper.finalize()
SyntaxError: helper not defined
```

---

Figure 15: Case B: Over-pruning the hallucinated anchor **get_config()** causes removal of a shared-scope helper, resulting in an unresolved reference (helper not defined).

## N   CSHS Generalization Across Models and Tasks

We assess the generalizability of CSHS beyond its original training context by evaluating its transfer performance across models and tasks.

