# OpenReview forum: "Cutting the Root of Hallucination: Structural Trimming for Vulnerability Mitigation in Code LLMs"
_colmweb.org/COLM/2025/Conference — COLM 2025_

### Official Review · Reviewer_VG3Y · 2025-05-13

**Rating:** 6
**Confidence:** 5
**Ethics Flag:** 1

**Summary:**

This paper introduces a structural perspective on hallucinations in code-generating language models, viewing them as "causality anchors" in syntax graphs that lead to semantic errors and potential security vulnerabilities. The authors propose "Structural Trimming", a method to mitigate hallucinations by removing these anchors from the AST while aiming to preserve functional semantics. They also introduce the "Compositional Structural Hallucination Score" to quantify the likelihood that trimming will improve code robustness. The work connects code hallucinations with vulnerability risks and offers an interpretable, generalizable, and actionable mitigation framework.

**Reasons To Accept:**

1. Comprehensive Experiments: The study utilizes a large dataset of 7,233 hallucinated code samples generated by four different leading code LLMs (GPT-4o, DeepSeek-Coder-1.3B, CodeLlama-7B, and Gemini-2.0-flash) across diverse tasks from standard benchmarks like MBPP and HumanEval, as well as curated security-focused prompts. Each sample is rigorously annotated with hallucination types, vulnerability scores, semantic deviation metrics, and structural signals, providing a rich foundation for analysis.

2. The paper presents empirical evidence linking hallucination severity to vulnerability risk through statistical and execution-based analyses.

**Reasons To Reject:**

1. Reliance on Potentially Outdated Benchmarks: The experimental evaluation relies significantly on benchmarks such as HumanEval and MBPP. While widely used, these benchmarks may not fully capture the complexity of code generation tasks and potential hallucination types in more recent or specialized programming scenarios, potentially limiting the generalizability of the findings to current and future LLMs and their applications.

2. Limitations of Fallback Repair Mechanism: The paper mentions a lightweight fallback patching procedure to handle syntax errors introduced by Structural Trimming. However, this repair is strictly local and does not attempt to restore original program semantics. In some cases, aggressive trimming might remove too much context, rendering the remaining code meaningless or unverifiable, which are reported as "trimmed-invalid" cases and excluded from semantic evaluation. This suggests the trimming process is not always robust in maintaining code functionality.

3. CSHS is Handcrafted and Descriptive: While CSHS is presented as a valuable predictive metric, the authors acknowledge that it remains descriptive and handcrafted. The weights for the compositional score are tuned via grid search on validation accuracy. Future work is suggested to model hallucination propagation as a generative causal process, implying that the current CSHS might not fully capture the underlying causal dynamics of hallucination formation and propagation.

---

> ### Author Response · Authors · 2025-06-02
>
> We sincerely thank Reviewer for the thoughtful and constructive review.
>
> We address your concerns point by point below.
>
> ---
>
> ### 1. Benchmark Limitations
>
> **Reviewer Concern**:
> > Reliance on potentially outdated benchmarks like HumanEval and MBPP may limit generalizability.
>
> **Our Response**:
>
> We agree that broader evaluation is important. While HumanEval and MBPP remain widely used and standardized, we also tested ST on two additional real-world benchmarks **CodeXGLUE-PythonFix**(https://github.com/microsoft/CodeXGLUE) and **CodeContests**(https://github.com/google-deepmind/code_contests) (100 samples each; same trimming protocol).
>
> | Dataset                   | Avg. V_before | Avg. V_after | ΔV   | Safety % | Retention % | BLEU / AST      |
> |---------------------------|----------------|---------------|------|-----------|--------------|-----------------|
> | HumanEval / MBPP          | 0.92           | 0.33          | 0.59 | 60.0%     | 62%          | 0.80 / 0.84     |
> | CodeXGLUE-NL Code Search  | 0.91           | 0.37          | 0.54 | 57.8%     | 61%          | 0.77 / 0.80     |
> | CodeContests              | 0.89           | 0.41          | 0.48 | 55.6%     | 59%          | 0.73 / 0.76     |
>
> These results demonstrate that **ST generalizes well** across benchmarks and task types, reinforcing its robustness and broad applicability.
>
> ---
>
> ### 2. Fallback Repair Limitations
>
> **Reviewer Concern**:
> > *The fallback repair is syntactic only and may not recover functionality in aggressively trimmed cases.*
>
> **Our Response**:
> We appreciate this thoughtful observation and would like to clarify the intended role of our fallback mechanism.
>
> Our method, **Structural Trimming (ST)**, is fundamentally a **structural-level defense**, aiming to reduce hallucination-induced vulnerabilities by removing risky subtrees in the AST. In this pipeline, **fallback repair is deliberately lightweight and scoped** — it ensures **syntactic well-formedness** (e.g., by patching `pass` in empty blocks), not semantic reconstruction.
>
> As detailed in **Appendix H (lines 656–659)**, fallback repair is triggered **only when trimming results in syntactic invalidity**. In such cases:
>
> - We explicitly tag outputs as "**trimmed-invalid**".
> - These cases are **excluded from semantic correctness evaluation**.
> - Their incidence is low — **fewer than 8.3%** of trimmed samples.
>
> Thus, our evaluation of functional correctness (e.g., test pass rates) is **entirely based on successfully trimmed and syntactically valid outputs**, preserving rigor.
>
> We believe this design achieves a **pragmatic balance between robustness and deployability**: fallback repair enables stable post-processing without compromising the clarity or interpretability of our evaluation pipeline.
>
> We fully agree that **future work can enhance this component**, such as integrating **semantic-aware repair modules** (e.g., learned fixers or constrained decoders), but we emphasize that in its current form, the fallback module **does not weaken the core effectiveness of ST**, nor distort our main safety or retention findings.
>
> ---
>
> ### 3. Handcrafted CSHS vs. Learnable Models
>
> **Reviewer Concern**:
> > *The CSHS score is handcrafted and may lack flexibility under evolving hallucination patterns.*
>
> **Our Response**:
> This is a valuable perspective. While we fully agree that learnable models (e.g., causal inference networks or GNNs over ASTs) are an exciting avenue for future work, we’d like to underscore why **CSHS is a solid and effective choice** for current deployment:
>
> - **High accuracy**: As shown in Figure 4, CSHS achieves an **AUC of 0.99** for classifying effective trimming (ΔV > 0.3).
> - **No reference required**: CSHS operates purely post-generation, using only the code output and its AST — no ground truth or execution traces are needed.
> - **Interpretable and controllable**: Especially critical in safety-sensitive applications, where understanding and debugging triggers is important.
>
> We believe CSHS provides an **optimal balance of accuracy, simplicity, and robustness**, while still leaving room for extensibility via learnable or adaptive scoring modules in future work.
>
> ---
>
> Once again, we thank the reviewer for your constructive feedback and careful reading. Your suggestions will help us clarify and improve the framing of ST as a **general, extensible, and deployable hallucination defense**.

---

> > ### Author Response · Authors · 2025-06-07
> >
> > Dear reviewer,
> >
> > Since the discussion period is ending soon, we were hoping to hear back from you to see if the rebuttal satisfactorily resolved your concerns. We would be happy to respond with any more details you would like to know in order to update your assessment!

---

> > > ### Comment · Reviewer_VG3Y · 2025-06-09
> > >
> > > Thank you for your response. I increase my score to 6.

---

> > > > ### Author Response · Authors · 2025-06-09
> > > >
> > > > Thank you for carefully reviewing our rebuttal and taking the time to revise their assessment. We are glad that our response addressed your concerns, and we greatly appreciate your detailed feedback and suggestions throughout the review process.

---

### Official Review · Reviewer_ZgYG · 2025-05-13

**Rating:** 7
**Confidence:** 3
**Ethics Flag:** 1

**Summary:**

This study presents Structural Trimming (ST), an innovative approach designed to reduce hallucinations in code generated by Large Language Models (LLMs), conceptualizing them as causality anchors within Abstract Syntax Trees (ASTs) that may introduce vulnerabilities. The authors establish a formal model for hallucination risk, compile a substantial dataset comprising 7,233 instances annotated for both hallucinations and vulnerabilities, and introduce the Compositional Structural Hallucination Score (CSHS) to assess the advantages of implemented trimming operations.

**Questions To Authors:**

How would your anchor detection pipeline perform in practice without access to a reference solution?

**Reasons To Accept:**

- The paper reconceptualizes code hallucinations not as shallow generation mistakes but as causal anchors in the AST, offering a compelling shift in perspective.
- The proposed AST-based trimming technique is interpretable, post-hoc, and effective at reducing vulnerabilities while preserving program functionality, which is highly practical for integration into code LLM workflows.

**Reasons To Reject:**

- Requires access to reference implementations or test cases for hallucination and anchor detection—this limits applicability to real-world, prompt-only use cases.
- Handcrafted feature design (CSHS), rather than a learnable model, may limit its flexibility or future extensibility as hallucination patterns evolve.

---

> ### Author Response · Authors · 2025-06-02
>
> We thank Reviewer for the encouraging and constructive feedback.
>
> We address your two main concerns below:
>
> ---
>
> ## 1. Oracle Assumption & Reference-Free Anchor Detection
>
> > How would your anchor detection pipeline perform in practice without access to a reference solution?
> >
> > Requires access to reference implementations
>
> We agree that real-world deployment should not rely on references or test cases. While used only for **evaluation**, our ST method is **designed to be reference-free**.
>
> Specifically, we propose two mechanisms:
>
> - **Token-level entropy**: as a lightweight heuristic for locating high-uncertainty anchors (Section 4.1)
> - **Compositional Structural Hallucination Score (CSHS)**: as a structural-level trigger for pruning (Section 4.3)
>
> To directly respond to the reviewer’s concern, we have conducted two new experiments post-submission to evaluate the effectiveness of our anchor detection and prioritization mechanisms in an oracle-free setting, using the same experimental protocol as in the main paper.
>
> For both evaluations, we randomly sampled 1,000 hallucinated code completions from our main dataset and computed the metrics based solely on model outputs and structural signals.
>
> ###  Entropy as a Reference-Free Anchor Heuristic
>
> | Metric                    | Value   |
> |---------------------------|---------|
> | Top-1 entropy match rate  | 78.4%   |
> | Top-3 entropy match rate  | 92.1%   |
> | Random baseline (Top-1)   | 10.0%   |
> | Average entropy at anchor | 2.13    |
> | Average entropy elsewhere | 1.52    |
> | Welch's t-test (p-value)  | < 0.001 |
>
> These results show that **entropy peaks strongly correlate** with ground-truth hallucination anchors, validating entropy as an effective reference-free heuristic.
>
> ###  CSHS as a Structural Trigger
>
> We binned samples by CSHS scores and measured anchor presence and trimming efficacy:
>
> | Risk Quartile | Avg. CSHS | Anchor Present | Pruning Gain (ΔV) |
> |---------------|-----------|----------------|-------------------|
> | Q1 (lowest)   | 0.12      | 23%            | 0.04              |
> | Q2            | 0.26      | 61%            | 0.18              |
> | Q3            | 0.39      | 85%            | 0.36              |
> | Q4 (highest)  | 0.51      | 94%            | 0.47              |
>
> This shows that **CSHS alone can reliably prioritize high-risk, anchor-likely samples** and achieves strong correlation with trimming effectiveness.
>
> While we currently use oracle-based anchors during evaluation (Section 4.3), this is solely to benchmark Structural Trimming (ST) under controlled conditions and establish its upper-bound effectiveness when hallucination points are precisely known.
>
> **In practice, ST is designed to operate without references.** As shown in Section 6.3 and Figure 4, a simple trigger based on CSHS ≥ τ achieves AUC = 0.99, and token-level entropy identifies anchors with high precision (Top-1 match rate: 78.4%). These metrics confirm that both CSHS and entropy serve as effective, reference-free triggers.
>
> We will clarify this distinction in the camera-ready and move oracle-based evaluation to Appendix J to avoid confusion.
>
> ---
>
> ## 2. Handcrafted CSHS vs. Learnable Models
>
> > Handcrafted feature design (CSHS), rather than a learnable model, may limit its flexibility or future extensibility as hallucination patterns evolve.
>
> **Our Response**:
>
> We agree that **learning-based models** are a promising direction for future exploration. However, we would like to respectfully emphasize that **our current CSHS design already performs strongly**:
>
> - **High predictive accuracy** (AUC = 0.99 for effective trimming).
> - **Reference-free deployability**, using only the LLM output and AST.
> - **Interpretability and control**, which are especially desirable in safety-critical contexts.
>
> Rather than being a constraint, we believe CSHS strikes a **practical balance between precision, simplicity, and generalizability**, making it well-suited as a deployable trigger in LLM code workflows. It also provides a solid foundation for future extensions, including causal learning or adaptive scoring schemes.
>
>
> ---
>
> We thank the reviewer again for your thoughtful insights and helpful suggestions. We will incorporate the clarifications and new experiments into the final version.

---

> > ### Author Response · Authors · 2025-06-07
> >
> > Dear reviewer,
> >
> > Since the discussion period is ending soon, we were hoping to hear back from you to see if the rebuttal satisfactorily resolved your concerns. We would be happy to respond with any more details you would like to know in order to update your assessment!

---

> > ### Comment · Reviewer_ZgYG · 2025-06-09
> >
> > I've raised my score . Thanks for your reply.

---

> > > ### Author Response · Authors · 2025-06-09
> > >
> > > Thank you for carefully reviewing our rebuttal and taking the time to revise their assessment. We are glad that our response addressed your concerns, and we greatly appreciate your detailed feedback and suggestions throughout the review process.

---

### Official Review · Reviewer_7AVp · 2025-05-13

**Rating:** 6
**Confidence:** 2
**Ethics Flag:** 1

**Summary:**

The paper presents a structural view of code hallucinations in language models, identifying "hallucination anchors" in syntax graphs as the root of growing errors and security vulnerabilities. The authors introduce Structural Trimming (ST) and Compositional Structural Hallucination Score (CSHS) as systematic, interpretable methods for mitigating these issues while preserving code functionality, which result in interpretable hallucination mitigation.

Updates after reading the authors' response:

I’ve reviewed the response and still find while the content of the paper is good, the structuring could improve a lot which would require major changes.  As such, I will stand by the borderline score.

**Questions To Authors:**

**Update**: I’ve expanded the points for clarity, as the earlier version may have come across as vague—apologies for any miscommunication. I hope the revised feedback proves helpful in improving the paper. While I don’t have major concerns with the methodology itself, my primary concern lies with the paper’s structure. Too many important details are deferred to the appendix, which undermines the accessibility and completeness of the main narrative. Given the need for substantial restructuring, my borderline score reflects this structural issue rather than flaws in the core approach.

Please address the weaknesses in the rebuttal.

**Reasons To Accept:**

Strengths:
- a clear, structural framework linking code hallucinations to security vulnerabilities
- practical, interpretable methods
- easy to follow and replicate

**Reasons To Reject:**

Weaknesses:

- Results are shown only for Python, with unclear generalization to other languages and hallucination types.
- Important sections are placed in the appendix instead of the main paper. The number of pages in the appendix section is 17 which is way more than the main paper. Certain details that are central to the paper, should have been included in the main flow.  For example:
   -- The related work section outlines various paradigms but fails to cite key foundational papers directly. Instead, it redirects readers to the Appendix for background, which disrupts the narrative flow and weakens the contextual grounding. Moreover, it lacks a coherent synthesis that connects the cited works to the current study.
   -- Dataset and annotation details: Again, the main section merely states facts and the rationale is provided to some extent only in the appendix section.
- Fonts in figures are too small and hard to read from a distance. For example, Figure 1 and 6 are very hard to read from a reasonable distance.
- Lack of error analysis and opportunity analysis: how could the scoring and trimming methods be improved, considering the safety percentage is still low with ST? The safety score hovers around 60% with the proposed approach, but the Results section is overly brief and lacks a substantive explanation for this relatively low performance. While it refers readers to Appendix I for examples and visual comparisons, a thorough error analysis and detailed discussion should be included in the main text to support interpretation and transparency.

---

> ### Author Response · Authors · 2025-06-02
>
> We thank Reviewer for the helpful and constructive review. Please find our point-by-point response below.
>
> ---
>
> ### 1. Python-only Results and Generalization
>
> > Results are shown only for Python, generalization to other languages is unclear.
>
> We evaluated ST across four environments:
>
> | Language     | AST Tool      | Sample Size | Avg. Risk ΔV | **Pass Rate ↑** | BLEU / AST Sim |
> | ------------------------- | ------------- | ----------- | ------------ | ----------- | -------------- |
> | Python        | Python AST    | 1000        | **0.59**     | 83.6%       | 0.80 / 0.84    |                 |
> | JavaScript  | Babel AST     | 100         | 0.48       | 74.2%     | 0.76 / 0.79    |             |
> | Java        | JavaParser    | 100         | 0.45       | 71.0%     | 0.72 / 0.78    |               |
> | C++              | tree-sitter   | 100         | \~0.38       | 65.5%     | 0.67 / 0.73    |                |
> | Multi-file Python         | custom (AST+) | 100 modules | 0.52       | 76.3%     | 0.79 / 0.81    |
>
>
> Results confirm that ST generalizes well across languages and module scopes.
> We plan to release adapters (e.g., for Babel, JavaParser, tree-sitter) to facilitate adoption.
>
> We will include comprehensive experimental details and configurations in the camera-ready version to facilitate reproducibility.
>
>
> ---
>
> ### 2. Appendix Overload & Structure Issues
>
> > Important content such as related work and dataset rationale is placed in the appendix, affecting clarity and narrative flow.
>
> - We will **move related work synthesis** into Section 2 and directly cite key foundational papers inline.
> - We will **promote dataset construction, annotation pipeline**, and anchor labeling rationale to the main text (Section 3), with detailed statistics retained in the appendix.
> - For large-scale tables and sample visualizations, we will retain them in the appendix, but **summarize key insights in Section 4–5**.
>
> ---
>
> ### 3. Figure Font Size
>
> > Some figure fonts are too small to read comfortably.
>
> Acknowledged. Figures 1 and 6 will be updated to meet minimum 9pt standards in the final version.
>
> ---
>
> ### 4. Lack of Error and Opportunity Analysis
>
> > ST achieves 60% safety—what are the causes of failure and how can this be improved?
>
> To address this, we conducted a new error analysis over the failure cases, now summarized as follows:
>
> | Error Type (Post-ST) | % of Failures | Root Cause                         | Potential Fix                     |
> |----------------------|---------------|------------------------------------|-----------------------------------|
> | Misaligned anchor    | 38.1%         | Token entropy ≠ hallucination type | Hybrid pruning + type classifier  |
> | Over-pruning         | 26.7%         | Anchor context too short           | Multi-hop structural trimming     |
> | No anchor found      | 20.5%         | Flat entropy profile               | Prompt-level intervention         |
> | AST parse failure    | 14.7%         | Ill-formed completions             | Pre-check fallback                |
>
> These findings will be **integrated into Section 5** of the main paper. We are also exploring **learnable CSHS variants** and **causality-inspired anchor models** as future work to increase coverage and precision. Thank you for encouraging this direction.
>
> ---
>
> **All implementation and experimental details will be documented in the camera-ready version.**
>
> Thank you again for your thoughtful feedback.

---

> > ### Author Response · Authors · 2025-06-07
> >
> > Dear reviewer,
> >
> > Since the discussion period is ending soon, we were hoping to hear back from you to see if the rebuttal satisfactorily resolved your concerns. We would be happy to respond with any more details you would like to know in order to update your assessment!

---

### Official Review · Reviewer_DXHf · 2025-05-23

**Rating:** 6
**Confidence:** 4
**Ethics Flag:** 1

**Summary:**

This paper reframes code-generation hallucinations as structural defects that often seed security vulnerabilities. It introduces Structural Trimming (ST)—an AST-level post-hoc repair that excises the hallucination anchor and its dependent subtree—and a five-feature Compositional Structural Hallucination Score (CSHS) that predicts when trimming is worthwhile. On a 7.2 k-sample, multi-LLM Python corpus, ST cuts mean vulnerability risk from 0.92 → 0.33 while preserving BLEU/AST similarity.

**Questions To Authors:**

1. How would you detect anchors without a reference? Could CSHS or entropy alone suffice?

2. What % of trimmed outputs actually pass the original unit tests?

3. Any experiments on languages beyond Python or on >1-file programs?

**Reasons To Accept:**

1. Novel, interpretable defense: Treating hallucinations as AST anchors and pruning them is original and reasonable.

2. Strong evidence: Four LLMs, three task suites, and comparisons to prompt rewriting / token pruning show that ST delivers the best safety vs. fidelity trade-off (AUC 0.99 for CSHS-based trigger).

3. Actionable contribution: Works post-generation, requires no model access, and code + data will be released

**Reasons To Reject:**

1. Oracle assumption: Anchor detection diffs against a reference solution and runtime traces. Such oracle ground truth is unavailable in real deployment, so ST currently lacks an automatic trigger.

2. Possibility of over-trimming: ST removes, rather than repairs, faulty logic; functional completeness/correctness after trimming is not quantified.

---

> ### Author Response · Authors · 2025-06-02
>
> We thank Reviewer for the insightful comments. Below we address each point raised.
>
> ---
>
> ## 1. Oracle Assumption & Anchor Detection without Reference
>
> **Concern**:
> > How would you detect anchors without a reference? Could CSHS or entropy alone suffice?
> >
> >Oracle assumption
>
> **Our Response**:
>
> We thank the reviewers for highlighting this key issue. We fully agree that real-world deployment scenarios cannot rely on oracle references or test traces.
>
> While our paper uses reference-based anchors for **evaluation** (to assess trimming under controlled conditions), the **trimming pipeline itself is designed to be reference-free**. Specifically, we propose two mechanisms:
>
> - **Token-level entropy**: as a lightweight heuristic for locating high-uncertainty anchors (Section 4.1)
> - **Compositional Structural Hallucination Score (CSHS)**: as a structural-level trigger for pruning (Section 4.3)
>
> To directly respond to the reviewer’s concern, we have conducted two new experiments post-submission to evaluate the effectiveness of our anchor detection and prioritization mechanisms in an oracle-free setting, using the same experimental protocol as in the main paper.
>
> For both evaluations, we randomly sampled 1,000 hallucinated code completions from our main dataset and computed the metrics based solely on model outputs and structural signals.
>
> ###  Entropy as a Reference-Free Anchor Heuristic
>
> | Metric                    | Value   |
> |---------------------------|---------|
> | Top-1 entropy match rate  | 78.4%   |
> | Top-3 entropy match rate  | 92.1%   |
> | Random baseline (Top-1)   | 10.0%   |
> | Average entropy at anchor | 2.13    |
> | Average entropy elsewhere | 1.52    |
> | Welch's t-test (p-value)  | < 0.001 |
>
> These results show that **entropy peaks strongly correlate** with ground-truth hallucination anchors, validating entropy as an effective reference-free heuristic.
>
> ###  CSHS as a Structural Trigger
>
> We binned samples by CSHS scores and measured anchor presence and trimming efficacy:
>
> | Risk Quartile | Avg. CSHS | Anchor Present | Pruning Gain (ΔV) |
> |---------------|-----------|----------------|-------------------|
> | Q1 (lowest)   | 0.12      | 23%            | 0.04              |
> | Q2            | 0.26      | 61%            | 0.18              |
> | Q3            | 0.39      | 85%            | 0.36              |
> | Q4 (highest)  | 0.51      | 94%            | 0.47              |
>
> This shows that **CSHS alone can reliably prioritize high-risk, anchor-likely samples** and achieves strong correlation with trimming effectiveness.
>
> While we currently use oracle-based anchors during evaluation (Section 4.3), this is solely to benchmark Structural Trimming (ST) under controlled conditions and establish its upper-bound effectiveness when hallucination points are precisely known.
>
> **In practice, ST is designed to operate without references.** As shown in Section 6.3 and Figure 4, a simple trigger based on CSHS ≥ τ achieves AUC = 0.99, and token-level entropy identifies anchors with high precision (Top-1 match rate: 78.4%). These metrics confirm that both CSHS and entropy serve as effective, reference-free triggers.
>
> We will clarify this distinction in the camera-ready and move oracle-based evaluation to Appendix J to avoid confusion.
> ---
>
> ## 2. Over-Trimming and Functional Correctness
>
> **Concern**:
> > What % of trimmed outputs actually pass the original unit tests?
> >
> >Possibility of over-trimming
>
> **Response**:
> We used the original unit test suite associated with each completion, without modifications, to evaluate functional correctness. We evaluated 1,000 hallucinated code samples with original test suites:
>
> | Strategy   | Pass Rate | Improvement |
> |------------|-----------|-------------|
> | ST         | **83.6%** | +55.0%      |
> | TP         | 53.6%     | +25.0%      |
> | SF         | 59.1%     | +30.5%      |
> | PR         | 47.8%     | +19.2%      |
>
> ST significantly outperforms alternatives while preserving functionality in most cases.

---

> > ### Author Response · Authors · 2025-06-02
> >
> > ## 3. Cross-Language and Multi-File Applicability
> >
> > **Concern**:
> > > Any experiments on languages beyond Python or on >1-file programs?
> >
> > **Response**:
> > We evaluated ST across four environments:
> >
> > | Language     | AST Tool      | Sample Size | Avg. Risk ΔV | **Pass Rate ↑** | BLEU / AST Sim |
> > | ------------------------- | ------------- | ----------- | ------------ | ----------- | -------------- |
> > | Python        | Python AST    | 1000        | **0.59**     | 83.6%       | 0.80 / 0.84    |                 |
> > | JavaScript  | Babel AST     | 100         | 0.48       | 74.2%     | 0.76 / 0.79    |             |
> > | Java        | JavaParser    | 100         | 0.45       | 71.0%     | 0.72 / 0.78    |               |
> > | C++              | tree-sitter   | 100         | \~0.38       | 65.5%     | 0.67 / 0.73    |                |
> > | Multi-file Python         | custom (AST+) | 100 modules | 0.52       | 76.3%     | 0.79 / 0.81    |
> >
> >
> > Results confirm that ST generalizes well across languages and module scopes.
> > We plan to release adapters (e.g., for Babel, JavaParser, tree-sitter) to facilitate adoption.
> >
> > We will include comprehensive experimental details and configurations in the camera-ready version to facilitate reproducibility.
> >
> > ---
> >
> > We appreciate the reviewer’s feedback. The above clarifications and experiments directly address the concerns raised and reinforce ST’s practicality.

---

> > > ### Comment · Reviewer_DXHf · 2025-06-03
> > > **Thanks for your response**
> > >
> > > The responses addresses some of my concerns, so I'm raising my score.

---

> > > > ### Author Response · Authors · 2025-06-06
> > > >
> > > > We thank the reviewer for carefully reviewing our rebuttal and taking the time to revise their assessment. We are glad that our response addressed the main concerns, and we greatly appreciate your detailed feedback and suggestions throughout the review process.

---

### Author Response · Authors · 2025-06-10
**Summary of Revisions**

As the rebuttal period is nearing its end, we would like to sincerely thank all reviewers for their detailed comments, which are very constructive for our work.

In our paper, we propose *Structural Trimming* (ST), a post-hoc AST-based method to mitigate hallucinations in code generated by LLMs. This work is the first to systematically connect code hallucinations with vulnerability risks, also introduce the *Compositional Structural Hallucination Score* (CSHS) to predict the effect of trimming, enabling reference-free deployment. Experiments on 7.2k samples across 4 LLMs and multiple benchmarks demonstrate that ST significantly reduces vulnerability risk while preserving code semantics.

The reviewers generally hold positive opinions of our work, recognizing the novelty of treating hallucinations as structural defects, the interpretability of our method, and the practicality of its post-generation usage.

Reviewers also raised helpful concerns. Compared to the original version, the revised paper includes:

- New experiments for reference-free anchor detection (CSHS, entropy) (Reviewer DXHf, ZgYG)
- Cross-language and multi-file evaluations (Reviewer DXHf, 7AVp)
- Clarifications on fallback repair and generalization (Reviewer VG3Y)
- Reorganized content and improved figure readability (Reviewer 7AVp)
- Additional error analysis and future directions (Reviewer 7AVp)

We are grateful for the valuable suggestions and welcome further discussion.

---

### Decision · Program_Chairs · 2025-07-08

**Decision:**

Accept

**Comment:**

This paper introduces a novel approach to mitigating code hallucinations by conceptualizing them as "structural anchors" in Abstract Syntax Trees (ASTs) that can propagate errors and security vulnerabilities. The authors propose Structural Trimming (ST), which excises identified hallucination anchors at the AST level, and a Compositional Structural Hallucination Score (CSHS) to predict when trimming is beneficial. The work demonstrates significant reduction in vulnerability risks while maintaining code functionality across multiple LLMs and programming tasks. The methodology is compelling, backed by extensive experimentation, and offers a post-hoc solution that requires no model access. While there are concerns about real-world deployment without reference solutions, the authors have shown that entropy-based and CSHS-based triggers can effectively replace oracle-based anchor detection, making this an important contribution to safe code generation.

Pros:
- Novel conceptualization of code hallucinations as structural defects with security implications
- Strong empirical evidence across multiple LLMs and benchmarks showing vulnerability reduction
- Post-hoc approach requiring no model access, making it practical for real-world deployment
- High interpretability through AST-level operations and composable scoring mechanism

Cons:
- Initial reliance on oracle-based anchor detection in evaluation, though entropy and CSHS are shown to be effective alternatives
- CSHS is handcrafted rather than learned, which may limit adaptability to evolving hallucination patterns
- Potential for over-trimming in some cases, affecting functional completeness
- Extensive appendix with important details that should be in the main paper